# Detection of Safe Passage for Trains at Rail Level Crossings Using Deep Learning

**DOI:** 10.3390/s21186281

**Published:** 2021-09-18

**Authors:** Teresa Pamuła, Wiesław Pamuła

**Affiliations:** Department of Transport Systems, Traffic Engineering and Logistics, Faculty of Transport and Aviation Engineering, Silesian University of Technology, Krasińskiego 8, 40-019 Katowice, Poland; Teresa.Pamula@polsl.pl

**Keywords:** rail level crossing, convolutional neural network, scene classification, traffic control

## Abstract

The detection of obstacles at rail level crossings (RLC) is an important task for ensuring the safety of train traffic. Traffic control systems require reliable sensors for determining the state of anRLC. Fusion of information from a number of sensors located at the site increases the capability for reacting to dangerous situations. One such source is video from monitoring cameras. This paper presents a method for processing video data, using deep learning, for the determination of the state of the area (region of interest—ROI) vital for a safe passage of the train. The proposed approach is validated using video surveillance material from a number of RLC sites in Poland. The films include 24/7 observations in all weather conditions and in all seasons of the year. Results show that the recall values reach 0.98 using significantly reduced processing resources. The solution can be used as an auxiliary source of signals for train control systems, together with other sensor data, and the fused dataset can meet railway safety standards.

## 1. Introduction

The safety of rail level crossings is the focus point of many activities of railway agencies [1]. Reports show that nearly a third of fatalities that are recorded in the course of operation of railways occur at RLCs [2]. Activities include changes in the construction of the sites and the introduction of sensing systems for surveillance of the rail tracks and road lanes.

Sensing systems may incorporate monitoring cameras, depth cameras, thermal cameras, radar, ultrasonic and LIDAR (light detection and ranging) sensors [3,4,5,6]. The devices provide a wide spectrum of information about the traffic at the RLC.

The problem is to reliably determine the state of the RLC using the least processing resources so as to enable an efficient integration of the solution with existing traffic control systems. Multiple sensing systems increase the reliability of operation when their outputs are combined. It is advantageous to utilize all sources of information for determining the state.

Monitoring cameras are an important source of data but are usually used only for recording incidents at RLCs. This gives a reason to work out a solution that uses the video stream contents as an input for determination of the RLC state. 

Safe passage of a train through an RLC is only possible when there are no obstacles on the tracks orin their vicinity. Atraffic control system requires a signal indicating the state of the RLC. The goal of processing the video stream is to detect obstacles or to classify the rail tracks as free of them. Changing light conditions strongly influence the efficiency of traditional image processing approaches for attaining this goal.

Deep-learning-based methods have proven successful for processing images recorded in highly changing light conditions [7]. This approach is used in this study to determine the state of the RLC. The goal is set to classify the area of the RLC using an appropriately tuned and trained CNN.

The main contributions of the authors are: design of a detection method for determining the state of the RLC without detecting obstacles in the area between gates,design of CNN for the classification of the content of image patches of the RLC area and thus, the determination of its occupancy instead of detecting obstacles in the area,validation of the design using a set of films monitoring the functioning of RLCs at different sites.

The reduction of the complexity of processing video data permits a hardware solution thatcan be integrated with camera setups. 

The rest of this paper is organized as follows. A review of related works is presented in Section 2. The next section introduces the proposed method of determining an empty RLC. Validation course and results are discussed in Section 4, and, lastly, conclusions and provisions for future work are presented

## 2. Related Works

Studies related to determining the state of RLC concentrate on the detection of incidents or obstacles using different sensor technologies. An incident or an obstacle signifies a busy RLC thatcoincides with the state of there being no way through the RLC for the train. 

Image-based approaches implement well-investigated image processing methods, such as background subtraction [8]. The crucial step in background subtraction is the design of the background model thatis updated with incoming video data. The complexity of the model determines the required resources for implementing it and its detection performance.

Authors in [9] report a five-frame background subtraction model with variance analysis adapted to the specifics of RLC sites. The model is tested using images of hazardous conditions at RLCs such as pedestrian crossing and stopped vehicles. Results show that the introduction of pixel intensity variance analysis makes it more robust to environmental noise.

An obstacle-detection system utilizing stereo cameras is presented in [10]. The design of the background model is based on colour independent component (ICA) technique. The idea of ICA is to restore statistically independent source signals, given only observed output signals without knowing the mixing matrix; in this case, these are the pixel values. Extraction of objects using background subtraction is done in the first step, and next, a carefully developed robust stereo-matching algorithm is used for localizing, in 3D, their positions at the RLC.

Observation of the RLC using thermal cameras greatly reduces the impact of changing light conditions, especially glare effects. This feature is highly desired but thermal sensors have much lower resolutions than standard camera sensors. The problem of designing a background model disappears as the thermal inertia of the environment is very high. Pavlovic et al. propose [5] a solution in this domain. Obstacleboundaries are obtained using Canny edge detector. The edge image is thresholded and a morphological closing is applied to close the gaps between edge segments. The result is a set of detected obstacles. The positions of these are estimated using the homography matrix of the camera. 

The problem of reducing the volume of video data for processing is addressed in [11]. The authors introduce an algorithm for dynamic multi-scale region processing for detecting intruders at railway surveillance sites. Downsampled images are used for finding candidate regions with intrusion targets, while the image data at full resolution in the candidate regions is used to identify the targets using background subtraction.

Background subtraction methods suffer from imperfections of modelling changing light conditions, and the results of determining the state of RLCs exposed to highly undetermined light changes are not always satisfactory. Artificial intelligence (AI) methods bring more robust solutions but at the cost of much higher processing power requirements. The search is for AI solutions optimized to the specifics of the posed problem. In this case, the domain of deep learning methods reasonably limits the scope of AI solutions.

Deep learning based methods are used in multiple transportation applications [12,13]. Deep learning algorithms use convolutional neural networks designed for processing structured arrays of data, such as images [14]. CNNs are widely adopted in the applications of video classification. A typical CNN consists of a convolutional layer whose goal is to extract pixel patterns throughout the training instances. It consists of multiple kernel filters that are applied to the contents of the image and transform the pixel values into higher-level descriptions [15,16,17].

The idea of using a CNN for classification exploits the higher=level descriptions for deriving the classification result. This implies that the input—image as a whole, is processed. When the contents of the image include a complex object with many features or a number of objects, a way of localizing the features or these objects is introduced to improve performance. 

Regions with CNN features (R-CNN) presented in [18] define a set of region proposals selected from the image using a selective search algorithm. Each region constitutes a CNN input. The extracted features are fed into a support vector machine (SVM) to classify the object in the region proposal.

The drawback of this approach is a large number of region proposals thatneed to be processed using CNNs so it requires manyresources to reach real-time operation. Proposed modifications Fast-RCNN [19] and Faster-RCNN [20] introduce modifications thatradically reduce the number of processing operations. The Fast-RCNN algorithm performs region selection on the convolutional feature map thatis the result of inputting the whole image to a CNN. The Faster-CNN omits the region selective search step and uses an additional network to predict region proposals. 

Region-processing-based approaches are hard to optimize because each step of the algorithms is usually tuned and trained separately. J. Redmon et al. proposed a (YOLO) you only look once algorithm [21] based on a single neural network thatpredicts bounding boxes and class probabilities directly from the whole image. The image is split into a grid where a number of bounding boxes is defined. The network outputs class probabilities for the boxes. Class probability above a threshold is selected and used to locate the objects within the image. Modifications of YOLO are being developed.

P. Sikora et al. propose YOLOv3 [22] for the classification and detection of the status of equipment mounted at RLC, that is barriers, traffic lights and warning lights. The processing is done using a GPU and reaches an average classification precision of over 96%. The authors test performance using several video recordings at different RLC sites. The solution is capable of working in real-time.

A lightweight CNN called BiMobileNet is presented in [23] for the classification of remote sensing images. Abilinear model is implemented, in which two parallel CNNs are used as feature extractors to obtain two deep features of the same image. MobileNetv2 is the backbone network for extracting features and provides data to the CNNs. The processing is done using a GPU, and classification accuracy reaches 94%.

The practical implementation of obstacle detection solutions at RLC sites needs to employ privacy-by-design and security-by-design best practices in order to secure all communication interfaces. This leads to the tight integration of processing resources with the data sources—cameras or sensors—forcing a search for less demanding processing algorithms. An artificial-intelligence-based surveillance system for railway crossing traffic (AISS4RCT) is a proposed system [24]. The YOLOv3-tiny model constitutes the base of the system and uses GPU acceleration boards placed inside camera modules observing the RLC site. This model achieves average recall (AR) value of 89%, processing 19 frames of video data per second.

## 3. Proposed Method

The proposed idea for detection of safe passage for trains changes the focus of determining the state of the RLC from extracting objects on the observation scene to the classification of the contents of the scene. Literature review reveals that object detection is predominantly used and with success, but at the same time the necessary processing resources are enormous. Changing the goal of the search for solutions redirects significantly the scope of study. 

The problem is phrased as follows: how best to efficiently determine one of the two states of the RLC? The two states are “with objects” and “safe passage for trains”. 

The work hypothesis is: an appropriately tuned and trained CNN processing a set of image patches of the RLC, from an observation camera, is adequate to determine the state of the RLC. The available processing resources are limited to embedded systems with GPUs so as to enable on-site processing. On-site processing, on safety grounds, forces condition monitoring of the systems. Signals of malfunctioning block the forwarding of classification results. 

Thanks to participation in R&D projects run by a RLC maintenance company, authors gained access to video surveillance material from a number of RLC sites in Poland. The films include 24/7 observations in all weather conditions and in all seasons of the year.

### 3.1. Database

The study uses a representative set of images recorded at one of the maintained RLC sites. The original video material with a resolution of 1920 × 1080 (FHD) is cut into frames for training the CNN and testing its performance. A random selection algorithm is used to obtain frames from a number of days of observation of the RLC at different times of the day and during different weather conditions. Figure 1 presents a sample of frames recorded at different times of day.

Two frame sets are prepared: one with objects, cars, passing trains, bicycles and pedestrians, the other with empty area between the gates of the RLC.

### 3.2. Determination of the RLC State

Limited processing resources determine the domain of the search for a solution. The reduction of data for processing is proposed and is performedin two ways. First, a ROI is defined, which covers only the area between the gates of the RLC. This area must be free of objects when the train approaches and passes through the RLC, irrespective of the gate states. Next, a square grid is introduced to reduce the input data for the CNN. The CNN processes input data from the square patches restricted by the grid. 

Figure 2 presents the RLC ROI marked in yellow (a) and an example with a car (b). The squares with the car have red edges. Two states of the ROI are defined: (1) safe passage for trains—the train can safely pass though, and all the squares are free of objects, (2) with objects–the safety of passage is endangered, and one or more squares contain objects or parts of objects.

The size of the grid squares is determined by the requirements of object detection and parameters of the observation cameras. The smallest objects thatmay disrupt the safe passage of the train are taken to be of the size of a cube with 0.3 [m] sides. Cameras are mounted on posts near the gates of the RLC so the field of view covers a quadrangle with sides no longer than 50 [m]. Taking into account the resolution of the camera, a square with 120 pixel sides is proposed. Figure 3 shows examples of patches thatare taken as inputs for training and testing the designed CNN.

The contents of the squares are processed using a tuned and trained CNN. It may be done serially or in parallel depending on the available processing device. GPU-based processing is advantageous as the CNN can be duplicated in the structure of the device.

The optimal architecture of the CNN is derived in an iterative manner by testing the classification performance of successive designs differing in the number of layers and by changing the filter parameters. The least complex solution is tantamount to the least computationally demanding, which is the goal of the design process.

The iteration begins with seven convolution layers and 32 filters of the size 7 × 7. The training process is done using data from the prepared frame sets. Most (60%) of the frames areused for training and the rest for evaluating the performance. Extracted squares from the RLC ROI for each of the frames are used as inputs. Each of the iterations is repeated a number of times to obtain a stable classification result. Gradually reducing the complexity of the network brings about the desired design.

ACNN with three convolution layers and two fully connected layers is proposed for the classification of the contents of the squares. Figure 4 presents the block diagram of the CNN. The first convolutional layer has 32 5 × 5 filters; the next layer has 16 3 × 3 filters, and the third has 16 3 × 3 filters. Downsampling of the convolution layers is performed using the max pool with 3 × 3 non-overlapping pooling regions. A rectified linear unit is used as a non-linear activation function for convolutional layers and for the fully connected layers. Local response normalization is used for the first two convolution layers. The first fully connected layer has 128 neurons. The output layer has 2 class-specific outputs.

The proposed classification method consists of the following steps, as shown in Figure 5b. Video data is acquired from the cameras located at the RLC. The video stream is cut into frames. The contents of the frames is masked by the ROI mask covering the area important for a safe passage of the train. The ROI consists of a number of square patches. Video data from these squares constitutes the input of the CNN. In the case of using GPU for processing, the CNN can be duplicated. Multiple CNNs significantly accelerate the processing of video data.

The results of the classification of the contents of each of the squares are collected, and the RLC state is calculated. When all the squares are empty—that is there are no objects or parts of objects in the observed area—the state of the RLC is determined to besafe for the passage of trains.

The proper parameters of the CNN are obtained in the process of training using frame sets as in Figure 5a. There are two sets defining the classification scope: one with objects or parts of objects and the second showing empty road and rail tracks. Video data comes from the site thatis the subject of classification. The square patches of the RLC ROI are used for training. A setup prepared for another site may be a useful start point for training and greatly reduces the training time.

### 3.3. Case Study

The validation of the proposed method is done using surveillance data recorded at one of the maintained RLC sites. The ROI as shown in Figure 2 contains 17 squares. The database for training and testing the CNN contains 1000 frames randomly extracted from the surveillance data. The frames are divided into two classes. Each class contains 500 elements. The training sets are prepared by cutting out squares in the defined ROI of the RLC frames. In all, 17,000 squares are used for training and testing.

## 4. Results and Discussion

The designed CNN is trained using 60% of the database. A number of training sessions aredone, and the best performing set of parameters is saved as the basis for the end construction of the CNN. The tests of the network are carried out using the rest of the database and the saved set of parameters. 

Figure 6 shows examples of RLC classification. Squares with small objects, as well as with large and multiple objects, are correctly classified. All examples signify that the state of the RLC does not allow a safe passage for a train.

Table 1 presents the RLC states’ determination results. Separate calculations are done for each of the states in order to determine which output is “safer” for controlling the rail traffic. Determining the state of RLC is closely related to securing the safety of its functioning. Monitoring cameras are usually additional sources of data for detection of risk threats.
(1)$$\begin{array}{l} precision=\frac{TP}{TP+FN}\\ recall=\frac{TP}{TP+FP}\\ F1=\frac{2\cdot precision\cdot recall}{precision+recall} \end{array}$$

The results are calculated using Equation (1), where TP is the number of correctly determined states of the RLC when the RLC is in state X; FP is the number of falsely detected states of the RLC when the RLC is in state X, and FN is the number of falsely determined states when the RLC is not in the state X.

Precision is the estimated probability that the RLC is determined as being in state X when the classifier works. Recall is the estimated probability that the RLC is determined as being in state X and RLC is in state X.

The determination result “safe passage for trains” has a higher recall value than “with objects”. This indicates that generating the signal for train control systems based on this class result is more justified. 

The value of F1- balanced F score is equal for both classes, which signifies a balanced performance of the CNN classifier.

The problem of classification of the squares can be transformed to classification of compressed representations of the contents of the squares. In the domain of deep learning, this is done using autoencoder neural networks. An autoencoder attempts to replicate its input at its output. Reducing the number of neurons in the hidden layer forces the autoencoder to learn a compressed representation of the input [25,26,27,28].

A two-layer autoencoder network is tested as a substitute of the CNN for comparison of performance. The first autoencoder transforms the vector of square pixels into a set of detail features. The autoencoder of the second layer reduces these features to a smaller set. The outputs are processed by a Softmax classification layer. 

The autoencoder neural network is trained and tested using the same database as in the case of the CNN. Several configurations are tested using a different number of neurons in the first autoencoder. The properties of this autoencoder determine the effectiveness of the network. 

The optimal structure consists of 400 neurons in the first layer and 100 in the second. The classification performance is presented in Table 2. An important difference must be noted that is poor precision indetecting “safe passage for trains”. This signifies difficulties in finding features for the classification of empty squares. 

The F1- balanced F score is much smaller than for the CNN, and it differs for the two classes. This performance indicates problems in distinguishing object parts and patches of rail and road. Such patches have similar features to some objects parts, which may be misleading.

A comparison of the results is done with a classification system using BoF (bag of features) and SVM (support vector machine), which is an approach prevailing in many image classification tasks not based on deep learning [29,30,31]. 

The image content is represented using SURF (speeded-up robust features) feature vectors. The feature vectors are clustered using K-means algorithm. The centroids of these clusters are the elements of BoF. The normalized histogram of the quantized features is the BoF model representation of the image content. SVM algorithm is used to determine the decision boundaries between the classes.

The same database is used for testing and assessing the classification performance. Table 3 presents asummary of the results. The classification performance is not significantly worse than in the case of the CNN. The precision of classification of the state “with objects” is 0.07 worse, whereas the recall value is 0.02 smaller. In the case of the BoF classification the signal for train control systems needs to be derived from the RLC state “with objects”.

The value of F1 score is equal for both classes, which signifies a balanced performance of the BoF classifier.

The determination of the state “safe passage for trains” is more sensitive to square classification errors, as not one out of the tens of squares covering the RLC may be classified as containing objects or parts.

The course of the validation and comparison with other solutions shows that some modifications to the determination of the RLC state can be introduced. The use of a fixed grid of squares leads to the processing of squares, which only in part are important for train passage. The introduction of overlapping squares, especially at the road edges, can bring some processing redundancy but will eliminate errors related to the wrong classification of content irrelevant to the RLC state. 

The camera provides a perspective view of the RLC, in which the nearest objects are mapped by a large number of pixels, whereas at the far end of the RLC, not much detail is visible. The size of the squares can vary depending on the placement in relation to the camera. This idea requires a number of CNNs adapted to the set of square sizes.

## 5. Conclusions

The determination of the state of the area between the gates of the RLC using deep learning, for signalling a safe passage for trains, proves successful. Determination results fall in the range of results noted in the literature review. The proposed solution using a CNN with only three convolution layers is much less complicated, especially in comparison with YOLO-based solutions thatcontain more than 50 neural network layers.

The proposed solution significantly reduces the required processing resources for detection of safe passage for trains. The solution can be used as an auxiliary source of signals for train control systems, together with other sensor data, and the fused dataset can meet the railway safety standards.

The solution may be useful for monitoring the state of road junctions for signalling risk situations in ITS systems. 

Future work will focus on efficient implementation of the method using GPU-based embedded processing systems integrated with observation camera modules [32].

## Figures and Tables

**Figure 1 sensors-21-06281-f001:**
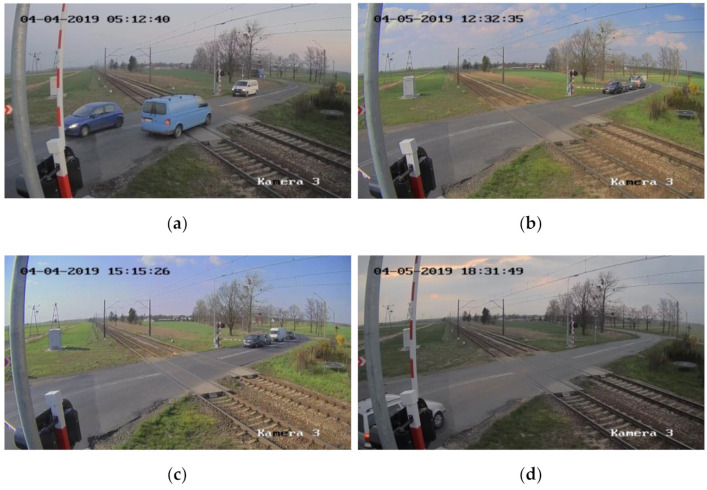
RLC at different times of day: (**a**) morning, (**b**) midday, (**c**) afternoon, (**d**) evening.

**Figure 2 sensors-21-06281-f002:**
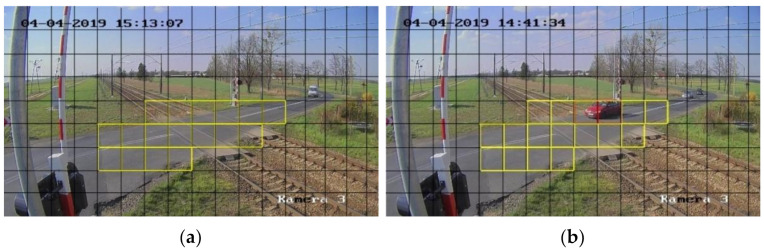
Image with the square grid: (**a**) safe passage for trains, (**b**) with objects.

**Figure 3 sensors-21-06281-f003:**
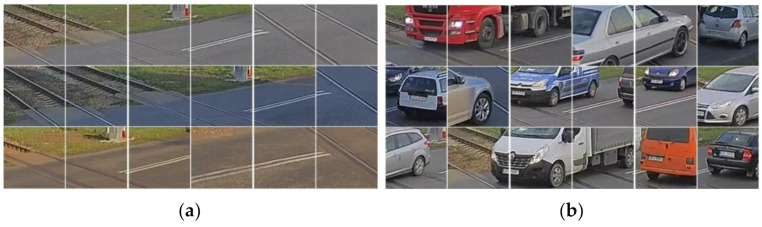
Square patches used for training and testing CNNs: (**a**) empty squares, (**b**) squares with objects or parts of objects.

**Figure 4 sensors-21-06281-f004:**
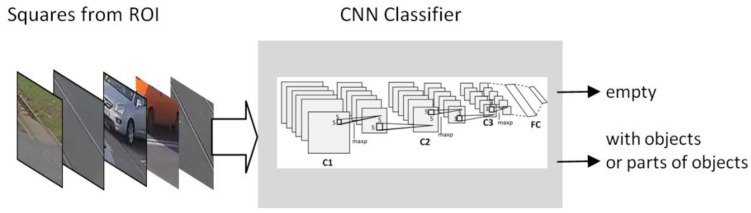
CNN classifier designed for the classification of the contents of the squares.

**Figure 5 sensors-21-06281-f005:**
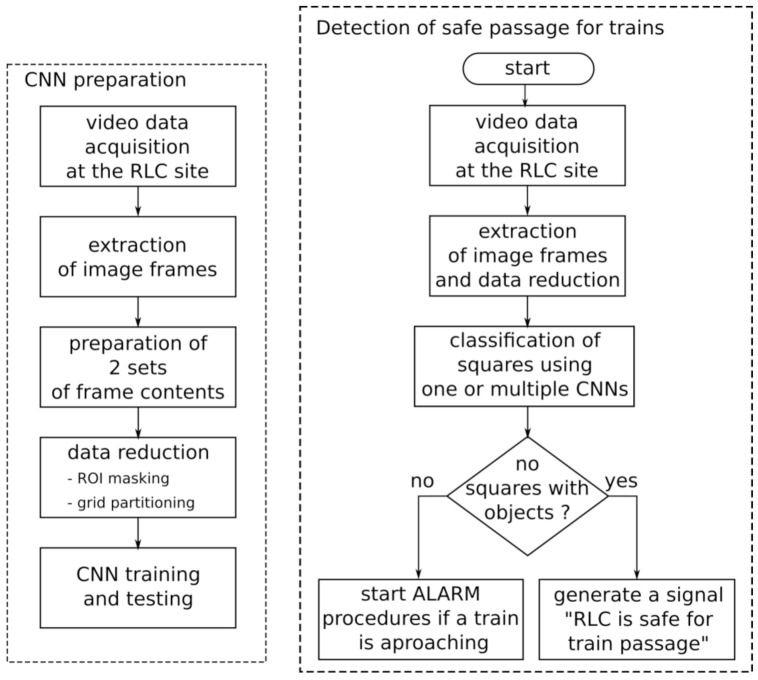
Block diagram of the method for classification of the state of the RLC: (**a**) preparation of the CNN, (**b**) determination of the state of the RLC.

**Figure 6 sensors-21-06281-f006:**
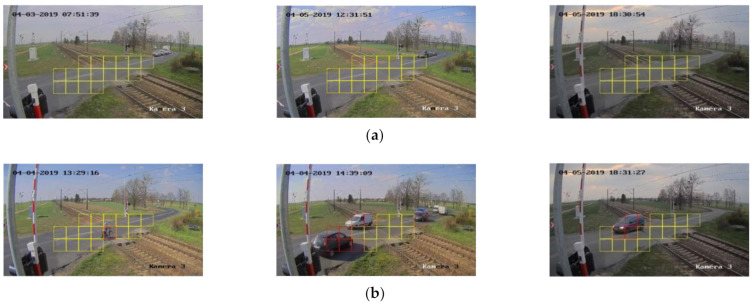
RLC state (**a**) “safe passage for trains”, (**b**) “with objects”. Red edges signify that the square contains objects.

**Table 1 sensors-21-06281-t001:** Determination of the RLC states using CNN.

RLC States	Precision	Recall	F1
with objects	0.98	0.94	0.96
safe passage for trains	0.94	0.98	0.96

**Table 2 sensors-21-06281-t002:** Determination of the RLC states using autoencoders.

RLC States	Precision	Recall	F1
with objects	0.99	0.79	0.84
safe passage for trains	0.73	0.99	0.88

**Table 3 sensors-21-06281-t003:** Determination of the RLC states using BoF.

RLC States	Precision	Recall	F1
with objects	0.91	0.96	0.93
safe passage for trains	0.96	0.91	0.93

## Data Availability

The data presented in this study are available on request from the corresponding author. The data are not publicly available due to privacy protection.

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
