# Peer review of "Detection of Safe Passage for Trains at Rail Level Crossings Using Deep Learning"

_sensors, 2021, doi:10.3390/s21186281_

Round 1
Reviewer 1 Report
I have a few comments on this paper, which proposes using computer vision and unreliable networks to determine if a Rail Level Crossing (RLC) is clear for the passage of trains. I thought the paper was very nicely presented. My comments are related to terminology used and
First, here in the US,Rail Level Crossing is an unknown term. We tend to use the term Grade Crossing. This was fairly quick to grasp, just surprising at first.
Second, you mention using this system for providing information to train crews about whether or not a crossing is clear. The question to consider here is can you provide relevant information to the engineer of the train when it is still possible to prevent a collision at an RLC.
At least in the US, I believe this is impractical. US freight trains are long and can take a considerable distance ( sometimes a mile or more, depending on speed and weight ) to slow to a stop. This means your system would need to predict whether or not the RLC will be occupied well before the train arrives.
On rail lines that are signaled, instructions from the signal system to slow the train to a stop will occur 4 or more miles from the desired stopping place. While the train slows during those 4 miles, the RLC may become obstructed or cleared multiple times
On unsignaled rail lines, there is no reliable way to ask the engineer to slow down before the RLC.
Author Response
Dear Reviewer,
thank you for reviewing our manuscript and for your supportive comment.
We have used European terminology as this is the domain of our research works.
The problem of providing information to the train crews about the state of the RLC is a difficult one. In Europe traffic management systems gain momentum, an integrated continental European Railway Traffic Management System (ERTMS) is being introduced. One of the elements of the system is to provide real time information for the train crew to safeguard the safety of train operations. These systems require sensor data from sites which constitute risks for the train operation. The output from our proposed system may be such a source of data for the systems.
We wish to express our appreciation for your in-depth comments.
Sincerely,
Teresa Pamuła, Wiesław Pamuła
Reviewer 2 Report
Abstract: Using deep learning for processing video data, the authors method detects obstacles at rail level crossings using video surveillance material from a number of RLC sites in Poland, in order to have a safe passage of the train. The safety problems at the rail level crossings, the reliably state of the RLC, the multiple sensing systems used, the goal and main contributions of the paper are described in chapter Introduction.
Chapter 2, based on the RLC background subtraction model with variance analysis, the model analyzes and detect any modification in RLC pattern (obstacle-detection system) due to pixel intensity variance analysis captured by stereo cameras, using background model analyses, based on independent color component (ICA) technique in order to detect and place into 3D position any RLC obstacle, using also an algorithm for dynamic multi-scale region processing, taking into account the large amount of data.
Assuming that the author model for determining the state of the RLC using Fast RCNN (where the region proposals are created using Selective Search) is an exponential increase in terms of speed, in terms of accuracy, there’s not much improvement. The chapter ends with practical implementation of obstacle detection solutions at RLC
Suggestion: Author must provide a short description of Convolution Neural Networks.
For future work, Region Proposal Network (RPN) must take into account in order to design this RPN to approximate ROIs which can be further processed.
Chapter 3 is very well structured and fundamented, describing the proposed idea for detection of safe passage for trains, determining the state of the RLC based on observation scene extracting objects in order to classify the contents of the area.
Subchapter 3.1. Database uses a representative set of recorded images for study and a random selection algorithm is used to obtain frames from observation of the RLC while subchapter 3.2. Determination of the RLC state (classification method flowcharts) refer to optimal architecture of the CNN, derived in an iterative manner by testing the classification performance of successive designs. Chapter 3 ends with subchapter 3.2. Case study, which presents the validation of the proposed method using surveillance data recorded only in one RLC.
For future work, the validation must take into account multiple RLCs, due to single or multiple railways.
Chapter 4, Results and Discussion, dealing with the training of the designed CNN, determining the state of RLC, comparison of performance between two-layer autoencoder network as a substitute of the CNN, a comparison of the results using BoF and SVM algorithms. In order to obtain a BoF descriptor, we need to extract a feature from the image, such as SIFT (Scale Invariant Feature Transform), SURF (Speeded Up Robust Features), and LBP (Local Binary Patterns), etc.
The paper ends with conclusions well formulated and bibliography.
Author Response
Dear Reviewer,
thank you for reviewing our manuscript and for your supportive comments.
“Suggestion: Author must provide a short description of Convolution Neural Networks”.
Thank you for the suggestion. We have decided to omit a section on CNN basics as our focus was to present the development of the application of CNNs. We include supplementary bibliography items to provide the background of CNN use.
The following suggestion for future work is very inspiring. Application of RPN for ROI approximation complicates the processing algorithm, but may enable some reduction in the total processing balance. Thank you for articulating this noteworthy idea we will pursue this approach in streamlining our sensor.
“For future work, the validation must take into account multiple RLCs, due to single or multiple railways.”
Indeed this suggestion is matched with our future works. We have video material for the validation using multiple RLCs, with single and multiple railway tracks and this is our future task. We consider this as very important in the course of developing a working sensor capable for providing data to railway control systems.
We wish to express our appreciation for your in-depth comments and suggestions.
Sincerely,
Teresa Pamuła, Wiesław Pamuła

Reviewer 3 Report
The paper deals with the detection of obstacles at rail level crossings (RLC), seen as a machine vision problem. The authors use deep learning to determine whether the passage is safe for the train to pass. The authors provide a good overview of the current bibliography and outline the benefits of their approach.
Author Response
Dear Reviewer,
thank you for reviewing our manuscript and for your supportive comment.
Sincerely,
Teresa Pamuła, Wiesław Pamuła
Reviewer 4 Report
The authors developed deep learning method for the detection of safe passage for trains at rail level crossings. The background and related works are well-described. The proposed method and the resuts are well-validated. The work by itself is systematic, and I recommend publication in current form.
Author Response

(The authors gave the same response as above.)
